# Identification of coexisting *Mfrp^rd6* and *Pde6b^rd10* mutations causing spontaneous retinal detachment in commercially available rd6 mice

Asaka Lee Shiozawa[1,2☯], Maika Hosoi Kobayashi[1], Yusuke Shiozawa[2], Yasuhiro Ikeda[3], Yoshitaka Miyagawa[4], Fumiki Okamoto[1], Mashito Sakai[4], Takashi Okada[5], Tsutomu Igarashi[4,6☯*]

**1** Department of Ophthalmology, Nippon Medical School, Sendagi, Bunkyo-ku, Tokyo, Japan, **2** Laboratory of Molecular Analysis, Nippon Medical School, Sendagi, Bunkyo-ku, Tokyo, Japan, **3** Faculty of Medicine, University of Miyazaki, Kihara, Kiyotake, Miyazaki City, Miyazaki, Japan, **4** Department of Biochemistry and Molecular Biology, Nippon Medical School, Sendagi, Bunkyo-ku, Tokyo, Japan, **5** Division of Molecular and Medical Genetics, Center for Gene and Cell Therapy, Institute of Medical Science, University of Tokyo, Shirokanedai, Minato-ku, Tokyo, Japan, **6** Department of Ophthalmology, Nippon Medical School Chiba Hokusoh Hospital, Kamakari, Inzai City, Chiba, Japan

☯ These authors contributed equally to this work.
* tutomu@nms.ac.jp

## Abstract

### Purpose

The rd6 mouse model, characterized by retinal degeneration due to an *Mfrp* mutation, has been widely studied. However, we identified a subset of rd6 mice that developed severe non-rhegmatogenous retinal detachment (rd6-RD), suggesting the presence of additional genetic factors. This study aimed to characterize the retinal phenotype of rd6-RD mice and identify potential causative genetic mutations.

### Methods

We performed optical coherence tomography, fundus imaging, electroretinography, and histological analysis to compare retinal structures and functions between rd6, rd6-RD, and C57BL/6J mice. Whole-genome sequencing was conducted to identify potential mutations associated with the retinal detachment phenotype.

### Results

Optical coherence tomography revealed retinal detachment in rd6-RD mice as early as 4 weeks old, with complete loss of the outer nuclear layer by 6 weeks. Fundus examination at 11 weeks showed pale fundi and narrowed, whitened retinal vessels in rd6-RD mice, distinct from rd6 mice. On electroretinography, rd6-RD mice displayed significantly diminished a- and b-wave amplitudes, with no detectable responses by 10 weeks. Histological analysis confirmed severe outer retinal

**Data availability statement:** All relevant data are within the manuscript and its Supporting information files.

**Funding:** This work was supported in part by a Grant-in-Aid for Scientific Research (c) (24K09301) from MEXT (Ministry of Education, Culture, Sports, Science and Technology), a research grant from Santen Pharmaceutical Co., Ltd., Teika Pharmaceutical Co., Ltd., and a research grant from Johnson & Johnson Surgical Vision, Inc. The funding organizations had no role in the design or conduct of this research.

**Competing interests:** The authors have declared that no competing interests exist.

degeneration and disappearance of the outer layers in rd6-RD mice. Whole-genome sequencing identified a missense R560C mutation in *Pde6b*, corresponding to the *Pde6b^(rd10)* mutation, in rd6-RD mice.

## Conclusions

A subset of rd6 mice exhibited severe retinal detachment and outer retinal degeneration, distinct from the previously characterized *Mfrp*-related phenotype. The identification of the *Pde6b^(rd10)* mutation suggests that these mice possess a dual-mutant genotype (*Mfrp^(rd6)* and *Pde6b^(rd10)*), exacerbating retinal degeneration. These findings highlight the importance of genetic verification in commercially available mouse models and provide new insights into the genetic complexity of inherited retinal degenerations.

---

## Introduction

Retinitis pigmentosa (RP) is a clinically and genetically heterogeneous group of inherited retinal disorders characterized by diffuse progressive dysfunction of rod photoreceptors, accompanied by degeneration of cone photoreceptors and the retinal pigment epithelium (RPE) [1]. Visual impairment typically manifests as night blindness and progressive visual field loss. The prevalence of RP ranges from 1:3000–1:5000 [2,3]. Effective treatment remains lacking and disease progression to blindness is currently inevitable.

In recent years, advances in the understanding of the genetic basis of RP have led to the development of various animal models [4]. Among these, the rd6 mouse is a naturally occurring model of autosomal recessive retinal degeneration, with slower disease progression compared to other models. This phenotype is caused by a 4-bp deletion in a splice donor site in *Mfrp* [5,6], a gene implicated in regulating the lipidome and transcription necessary for photoreceptor function [7,8]. Degeneration begins at 3–4 weeks of age and progresses slowly over approximately 16 months, affecting both rod and cone cells and causing gradual photoreceptor loss. The rd6 mouse model provides a valuable tool for studying the mechanisms underlying retinal degeneration and for developing potential therapies for this devastating condition [8,9]. Moreover, *Mfrp* mutations in humans have been associated with microphthalmia, and cystic macular detachment, in addition to *Mfrp*-associated retinal dystrophy [10].

In this study, we observed a subset of rd6 mice exhibiting early-onset retinal detachment, a feature not commonly reported in typical rd6 mice. To investigate the underlying causes, we compared rd6 mice showing retinal detachment (rd6-RD) to regular rd6 mice, with the aim of identifying factors contributing to this phenotype.

## Methods

### Animal husbandry

B6.C3Ga-*Mfrp^(rd6)*/J (rd6) mice [5,6] were obtained from Jackson Laboratory (Bar Harbor, ME), while C57BL/6J mice were purchased from CLEA Japan (Tokyo, Japan) as

controls. Mice were maintained under a 14-h/10-h light/dark cycle at a constant 25°C with ad libitum access to food and water. All animal procedures were approved by the Experimental Ethical Review Committee of Nippon Medical School (approval nos. H28-049 and 2021−024) and adhered to the ARVO Statement for the Use of Animals in Ophthalmic and Vision Research. Mice were treated with an intraperitoneal administration of a combination of ketamine (Daiichi Sankyo Co., Ltd., Tokyo, Japan) and xylazine (Bayer Medical, Ltd., Tokyo, Japan) at 100 and 10 mg/kg body weight, respectively. Exsanguination was performed under deep anesthesia induced by ketamine/xylazine as the method of euthanasia.

## Fundus examination

Color fundus images were obtained using a GENESIS-Df handheld fundus camera (Kowa, Aichi, Japan) with a 90D lens following anesthesia and mydriasis. Imaging was performed at 11 weeks of age.

## Imaging with OCT

OCT was conducted as described in our previous study [9]. Briefly, images were acquired using a Cirrus HD-OCT Model 4000 (Carl Zeiss, Oberkochen, Germany) with a customized 90D lens adaptor. Image resolution was 500 pixels (height) × 750 pixels (width). Average outer retinal thickness (between the outer plexiform layer and RPE) was measured at 200 pixels from the optic nerve head using Adobe Photoshop (Adobe Inc., San Jose, CA) and GIMP 2 (https://www.gimp.org/). C57BL/6J (n = 8), rd6 (n = 12), and rd6-RD (n = 8) eyes were analyzed at postnatal weeks 4, 6, and 11.

## Electroretinography (ERG)

Visual function was assessed via full-field scotopic ERG as previously described [11]. Mice were dark-adapted overnight and anesthetized with intraperitoneal injection of a ketamine/xylazine cocktail. Dark-adapted ERG responses were recorded using a hemisphere full-field stimulator for rodents (MAYO Corporation, Aichi, Japan), LS-W photostimulator (MAYO Corporation), and PowerLab 2/26 (ADInstruments, Sydney, Australia) as an A/D converter with Bio Amp ML132 amplifiers (ADInstruments). ERG was performed at 5 and 10 weeks of age using stimulation intensities of 0.02 and 2 cd·s/m$^2$.

## Histopathology

Mice were anesthetized and perfused with phosphate-buffered saline (PBS), followed by 4% paraformaldehyde in 0.1 M phosphate buffer. Eyes were enucleated, the anterior segments were removed, and the remaining eye cups were post-fixed for two days in 4% paraformaldehyde at 4°C. Samples were then dehydrated, cleared in xylene, and embedded in paraffin. Sections (4-µm thick) were cut along the vertical meridian through the optic nerve head using a microtome and stained with hematoxylin and eosin. Images were captured using a BX60 microscope (Olympus Corporation, Tokyo, Japan). Retinal thickness was measured in an area 500 µm from the optic disc, with measurements taken from three different sections and analyzed using Photoshop (Adobe Inc., San Jose, CA) and GIMP 2 (https://www.gimp.org/).

## Genetic analysis

**Deep sequencing.** All 18 exons of *Prkcq* were amplified via polymerase chain reaction using primers containing a NotI linker (S1 Table). Pooled polymerase chain reaction products were purified, digested with NotI, and ligated. Libraries were constructed using the NEBNext Ultra II DNA Library Prep Kit (New England Biolabs, Ipswich, MA). Sequencing was conducted with an Illumina MiSeq platform using a 150-bp paired-end protocol.

**Whole-genome sequencing.** Genomic DNA (100 ng) was processed using the TruSeq Nano DNA Library Preparation Kit (Illumina, San Diego, CA) and sequenced on a NovaSeq platform with a 150-bp paired-end protocol. Sequencing reads were aligned to the mm10 reference genome using Isaac aligner [12]. Variants were identified using Isaac Variant

Caller and filtered based on genomic annotations from SnpEff (version 4.1) developed by Pablo Cingolani. The detailed protocol is provided in the S1 File.

## Statistical analysis

All results are expressed as mean±standard error of the mean. Comparisons between control and experimental groups were performed using One-way analysis of variance (ANOVA) and post-hoc Tukey's honestly significant difference (Tukey's HSD) test. Statistical analysis was conducted using R version 4.2.2 (The R Foundation for Statistical Computing, Vienna, Austria), with values of $p < 0.05$ considered statistically significant.

## Results

### Retinal detachments observed in rd6 mice using OCT imaging

First, all mice used in this study were genotyped for $Mfrp^{rd6}$, which confirmed the presence of the homozygous $Mfrp^{rd6}$ mutation. In C57BL/6J mice, all retinal layers were clearly distinguishable (Fig 1a–1c), with a well-defined ellipsoid zone (EZ) line. In contrast, rd6 mice displayed an indistinct EZ line, although the outer nuclear layer (ONL) remained visible (Fig 1d–1f). Among rd6 mice, some exhibited retinal detachment as early as 4 weeks of age, accompanied by a significantly thinned ONL (Fig 2a). By 6 weeks, retinal detachment had progressed and the ONL was no longer detectable (Fig 2b). In this study, we classified mice with confirmed retinal detachment by 6 weeks as the retinal detachment group (rd6-RD). At 11 weeks, retinal detachment became more extensive (Fig 2c).

### Severe thinning of the outer retinal layers in rd6-RD mice

A comparison of magnified retinal images revealed significant thinning of the outer retinal layers in rd6-RD mice (Fig 3a). The relative thickness of the outer retinal layer was quantified among groups (S2 Table). Setting the mean value for

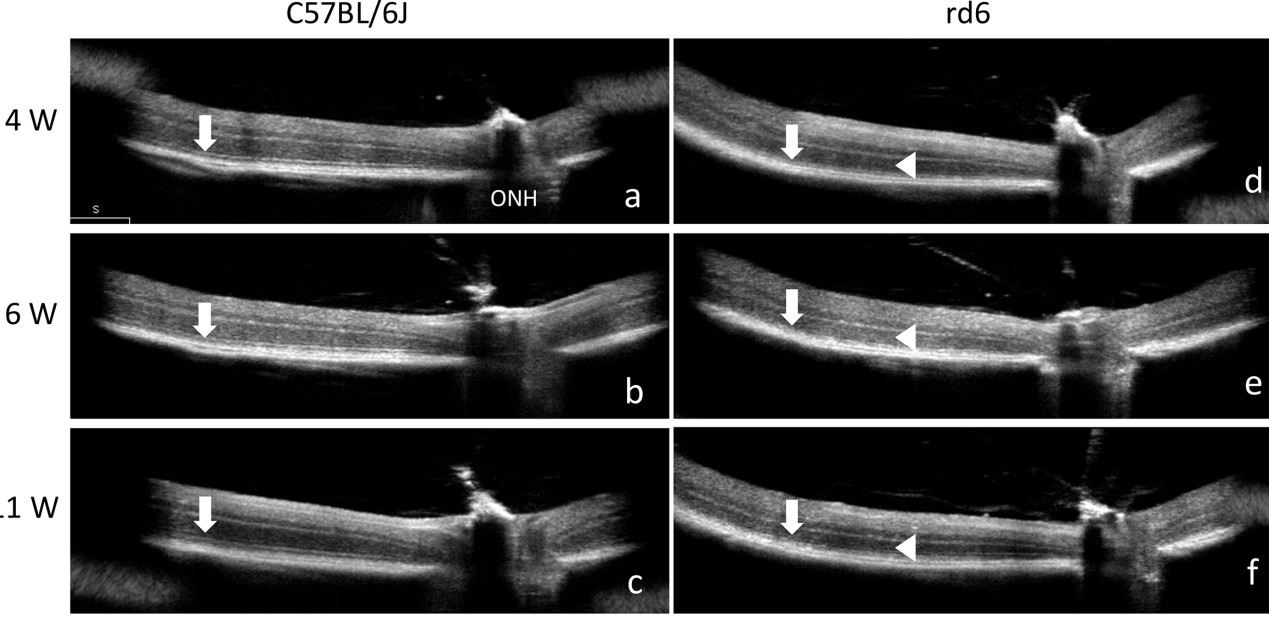

**Fig 1. OCT of the retina in C57BL/6J and rd6 mice.** OCT performed at 4 weeks (W) **(a)**, 6W **(b)**, and 11W **(c)** in C57BL/6J mice, and at 4W **(d)**, 6W **(e)**, and 11W **(f)** in rd6 mice. In C57BL/6J mice, the ellipsoid zone (EZ) line is clearly observed (a–c, arrow). In rd6 mice, the EZ line appears indistinct (d–f, arrow), but the outer nuclear layer (ONL) remains visible (d–f, arrowhead). ONH: optic nerve head.

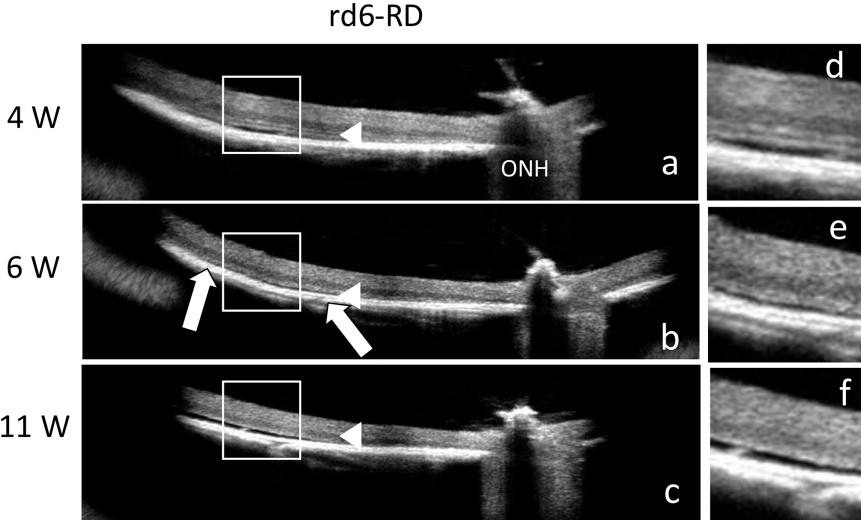

**Fig 2. OCT of the retina in rd6-RD mice.** OCT performed at 4W **(a)**, 6W **(b)**, and 11W **(c)** in rd6-RD mice, which developed retinal detachment. Magnified regions are indicated by squares **(d–f)**. Retinal detachment is observed as early as 4 weeks **(d)**, with a markedly thin ONL (a, arrow). By 6 weeks, retinal detachment has expanded (b, between arrows) and the ONL is no longer detectable (b, arrowhead). By 11 weeks, retinal detachment has further progressed **(f)**. ONH: optic nerve head.

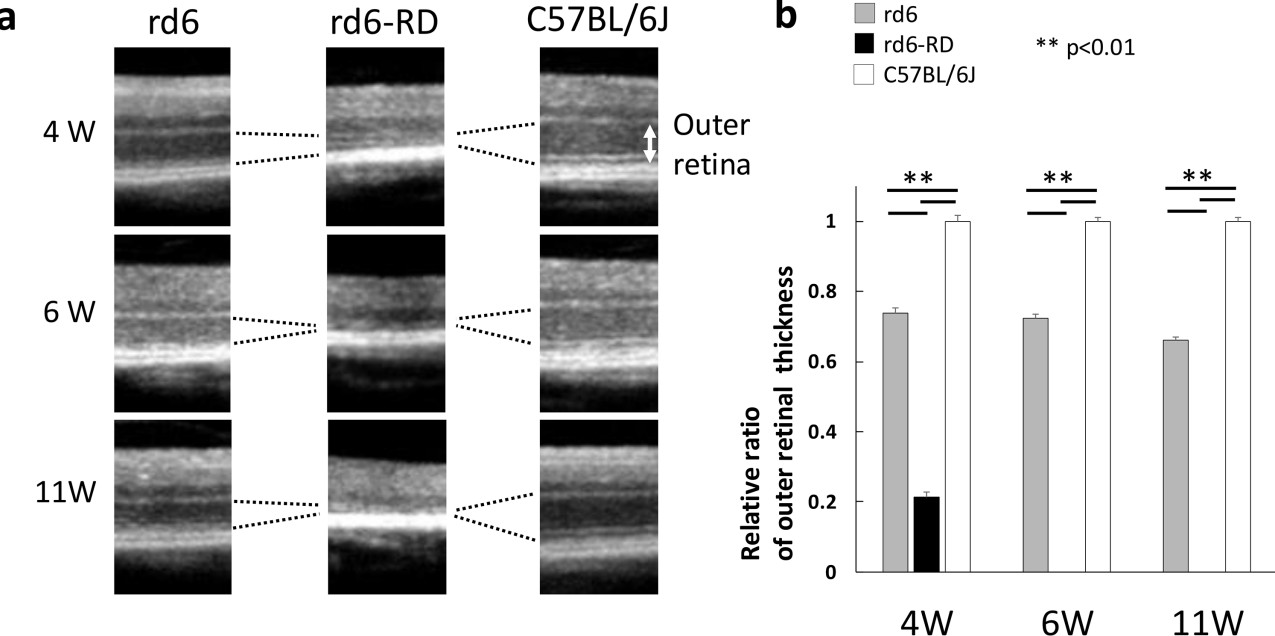

**Fig 3. Magnified OCT and relative outer retinal thickness in rd6, rd6-RD, and C57BL/6J mice.** **(a)** Magnified OCT images of the retina in rd6, rd6-RD, and C57BL/6J mice. Double arrows indicate the outer retinal layer. **(b)** Relative outer retinal thickness in rd6 (n = 12), rd6-RD (n = 8), and C57BL/6J (n = 8) mice. Data are presented as mean ± SEM. **: $p < 0.01$ (One-way ANOVA and Tukey's HSD test).

C57BL/6J controls (n = 8) as 1, relative thicknesses for rd6 mice (n = 12) were 0.739 ± 0.014 at 4 weeks, 0.725 ± 0.012 at 6 weeks, and 0.661 ± 0.008 at 11 weeks. In contrast, rd6-RD mice (n = 8) exhibited a value of 0.226 ± 0.015 at 4 weeks, with complete outer retinal loss thereafter (Fig 3b). Raw images of OCT are shown in S2 Fig.

### Fundoscopic abnormalities in rd6-RD mice

To further characterize the phenotype, color fundus images were captured at 11 weeks of age. Consistent with previous reports [5], rd6 mice exhibited multiple white spots of uniform size and shape (Fig 4b). However, while similar white spots were observed in rd6-RD mice, the fundi appeared pale and retinal vessels were narrowed and whitened, unlike in rd6 mice (Fig 4c).

### Severely diminished ERG responses in rd6-RD mice

To evaluate visual function, ERG recordings were obtained from C57BL/6J (n = 8), rd6 (n = 12), and rd6-RD (n = 16) mice (S2 Table). While rd6 mice exhibited significantly smaller a- and b-wave amplitudes compared to C57BL/6J controls, rd6-RD mice demonstrated the weakest responses at 5 weeks, with no detectable signals at 10 weeks under both dim and strong light intensities (Fig 5).

### Histological analysis reveals severe outer retinal degeneration in rd6-RD mice

To further assess the pathological condition of rd6-RD, paraffin-embedded retinal sections were stained with hematoxylin and eosin. The ONL of rd6 mice was significantly thinner than that of wild-type mice, but the outer retina of rd6-RD was drastically reduced, with no identifiable outer layers under the microscope (Fig 6a–6c). Mean outer retinal thickness was 86.5 ± 6.44 µm in C57BL/6J controls, 51.8 ± 2.2 µm in rd6 mice, and only 6.63 ± 0.74 µm in rd6-RD mice (S2 Table).

### Identification of the rd10 mutation through whole-genome sequencing

Given the severity of retinal degeneration observed in rd6-RD mice, we hypothesized the presence of an additional genetic mutation beyond *Mfrp* mutation. We first sequenced all the exons of *Prkcq*, of which mutations were known to be

| C57BL/6J | rd6 | rd6-RD |
|:---:|:---:|:---:|

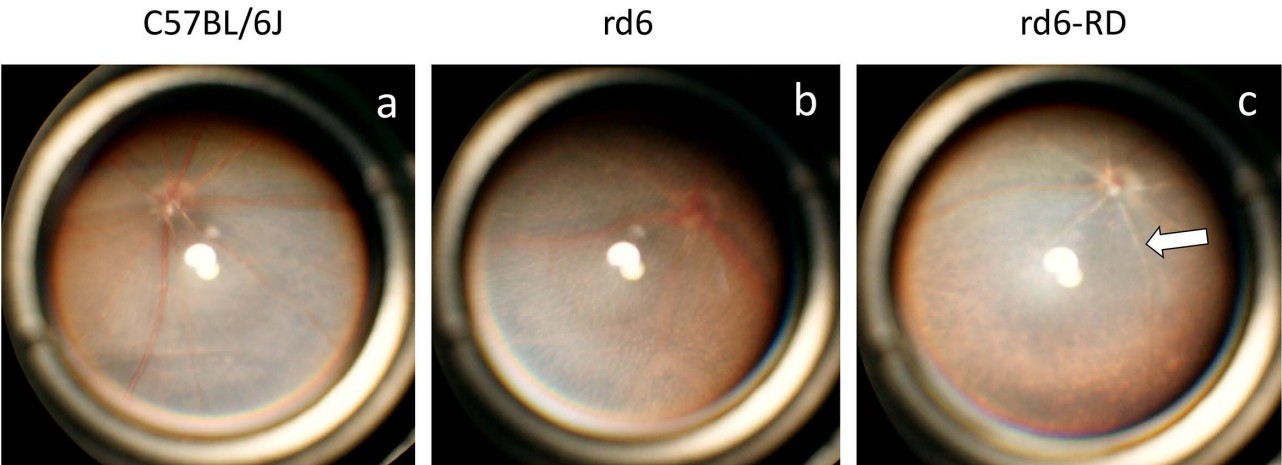

**Fig 4. Fundus images of rd6, rd6-RD, and C57BL/6J mice.** Fundus images captured at 11 weeks old. Multiple white spots of uniform size and shape **(b)** are seen in rd6 mice, consistent with previous reports. Similarly, rd6-RD mice **(c)** display white spots, but also exhibit pale fundi and narrowed, whitened retinal vessels, unlike rd6 mice (c, arrow).

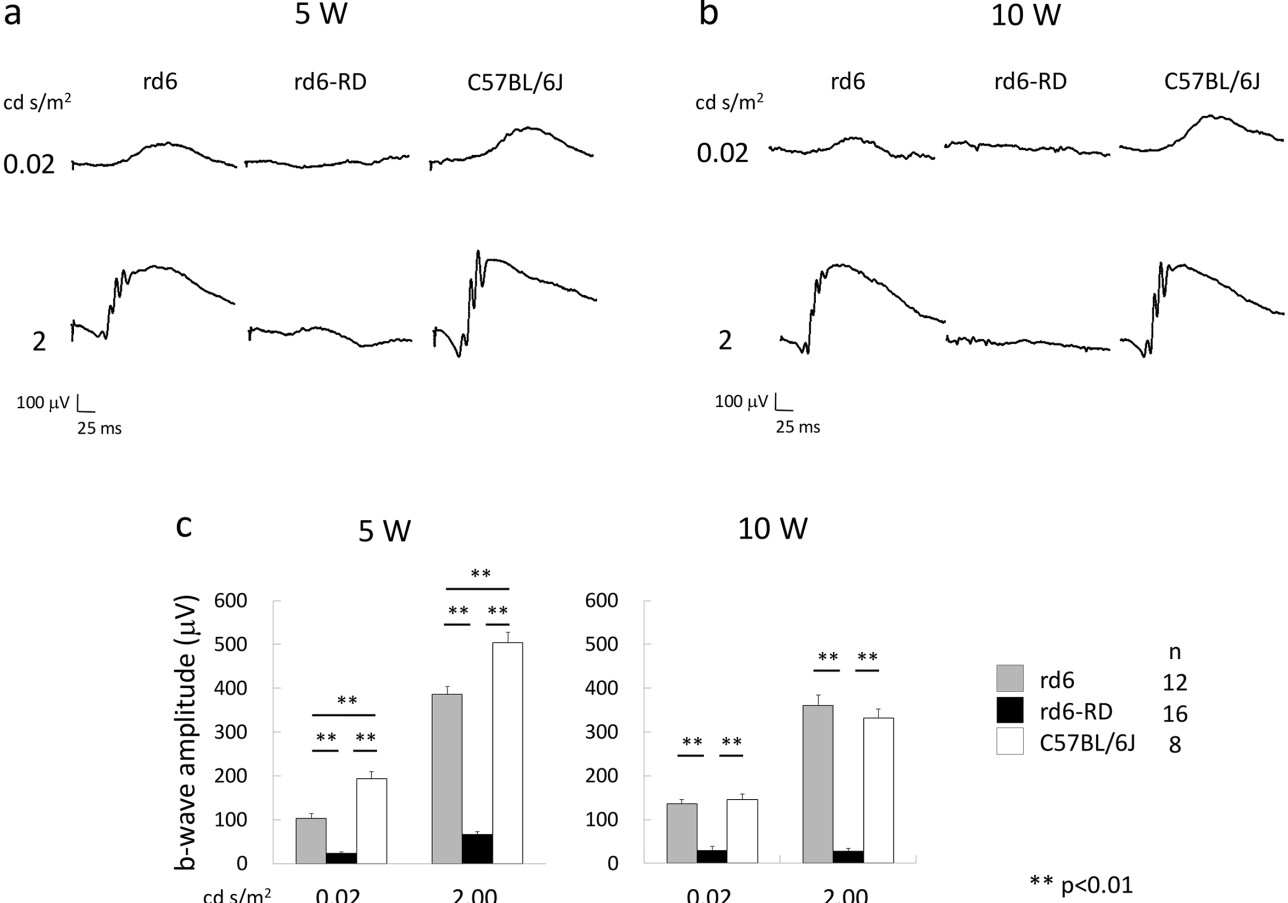

**Fig 5. ERG response in rd6, rd6-RD, and C57BL/6J mice.** To assess retinal function, scotopic ERG is performed to measure rod responses and mixed rod-cone responses under 0.02 cd·s/m² and 2 cd·s/m² stimuli at 5W **(a)** and 10W **(b)**. Under both dim and strong light intensities, rd6-RD mice exhibit the weakest responses at 5 W among the three groups and no detectable signals at 10 W **(c)**. Data are presented as mean ± SEM. **: p < 0.01 (One-way ANOVA and Tukey's HSD test).

associated with retinal detachment [13]. A comparison between a mouse exhibiting retinal detachment (237; Mouse identification number) and its unaffected littermate (236) revealed no mutations in *Prkcq*.

Next, whole-genome sequencing was performed on rd6-RD (with confirmed retinal detachment) and rd6 (without retinal detachment) mice to identify potential genetic variants. The analysis focused on a single family in which the father (294), four affected sons (408, 409, 410, 411), and one affected daughter (381) exhibited retinal detachment, whereas the mother (367), two unaffected sons (407, 412), and one unaffected daughter (382) did not (Fig 7a).

The mean coverage of whole-genome sequencing was 30.8× with 89.5% of the entire genome analyzed with ≥20 independent reads on average (S1Fig). In total, 176,432 variants were identified. We hypothesized two genetic models of inheritance:

1. rd6 mice are heterozygous, whereas rd6-RD mice are homozygous for the mutation.

2. rd6 mice are wild-type, whereas rd6-RD mice are heterozygous.

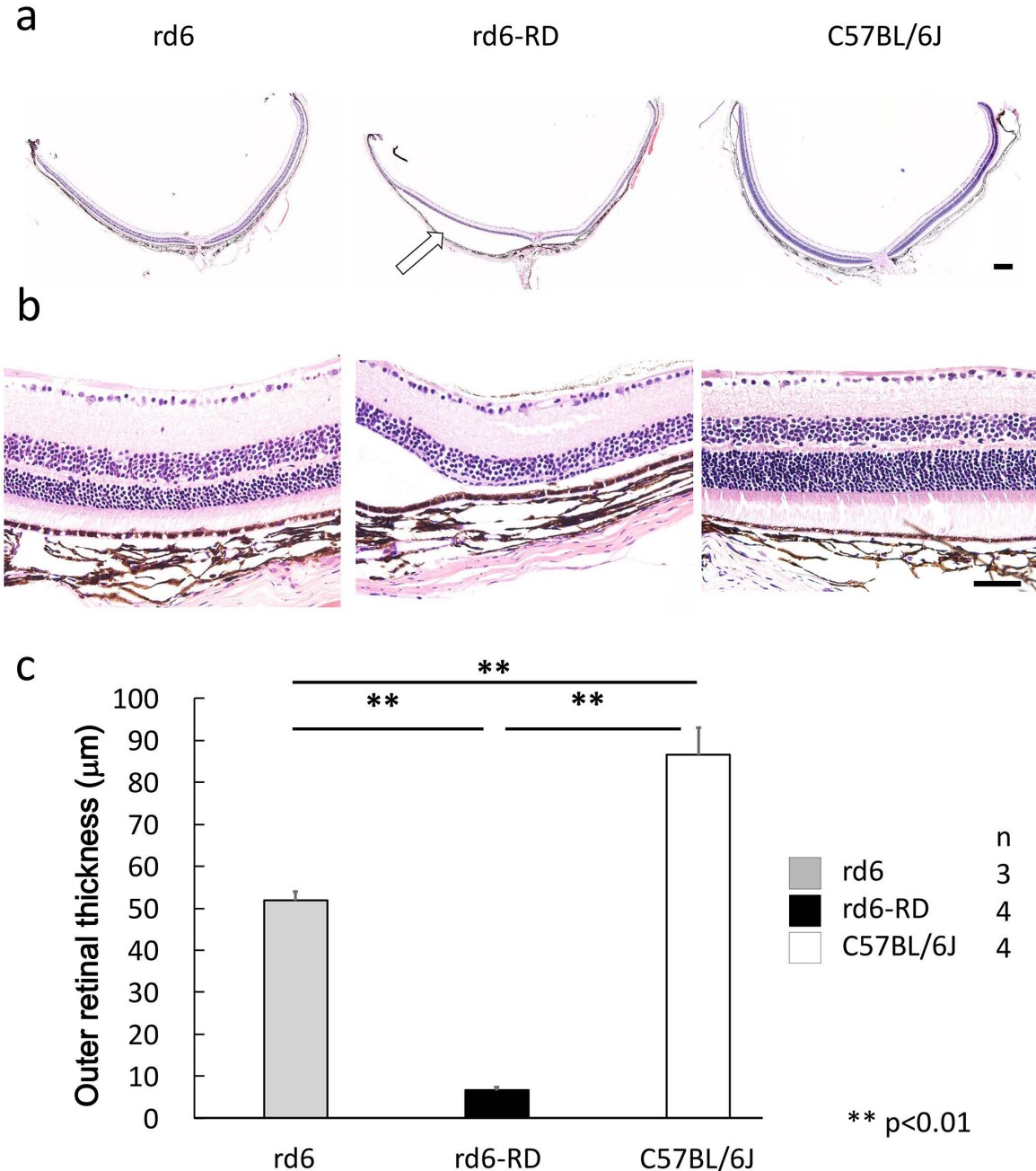

**Fig 6. Histological analysis and outer retinal thickness in rd6, rd6-RD, and C57BL/6J mice. (a)** Histological section of the posterior eye. Quantitative comparison of outer retinal thickness among rd6, rd6-RD, and C57BL/6J mice. Arrows indicate retinal detachment. **(b)** Magnified retinal section from each group. **(c)** The outer retinal layer is significantly thinner in rd6 mice than in C57BL/6J mice, while the outer layer has completely disappeared in rd6-RD mice. Data are presented as mean±SEM (S2 Table). **: p<0.01 (One-way ANOVA and Tukey's HSD test). Scale bars: 200 µm in (a); 50 µm in (b).

a

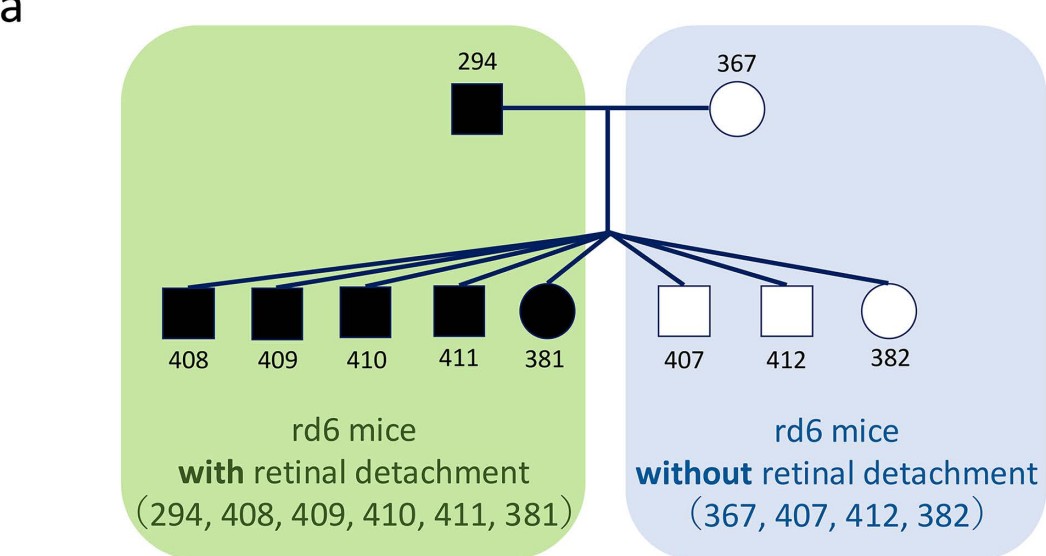

b

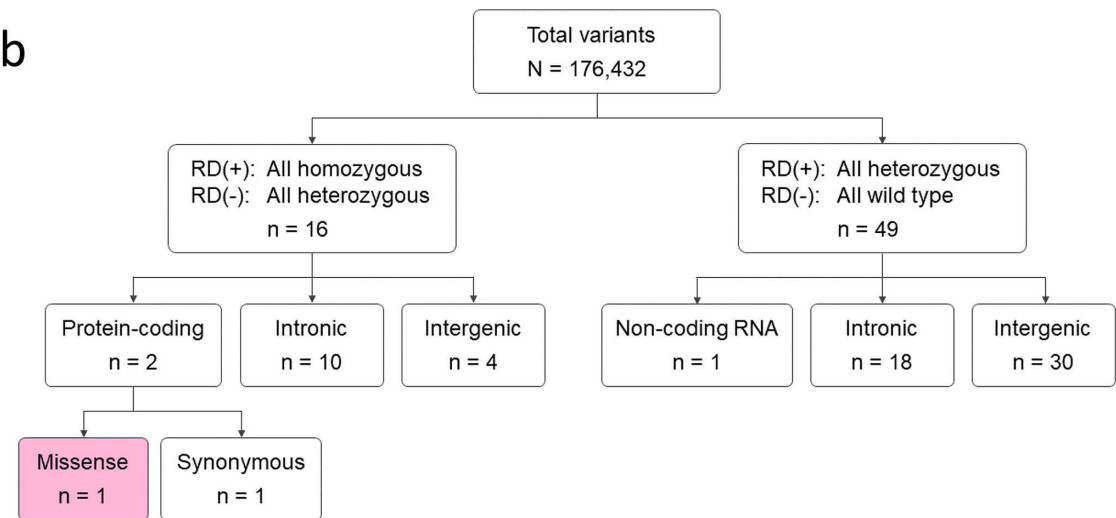

**Fig 7. Identification of the rd10 mutation via whole-genome sequencing. (a)** Pedigree of rd6 mice selected for whole-genome sequencing analysis. Males are represented by squares, and females by circles. Black-filled symbols indicate rd6 mice with retinal detachment. **(b)** Flowchart of variant filtering. RD(+): rd6 mice with retinal detachment; RD(-): rd6 mice without retinal detachment.

Sixteen variants fit the first inheritance pattern (homozygous mutation in rd6-RD), and 49 variants fit the second pattern (heterozygous mutation in rd6-RD). None of the variants fitting the second pattern were protein-coding (Fig 7b). Among the 16 variants fitting the first pattern, 2 were protein-coding, 10 were intronic, and 4 were intergenic (Fig 7b). Notably, one of the two protein-coding variants was a missense mutation in *Pde6b* (c.1678C>T, p.R560C) corresponding to the known *Pde6b*^rd10 mutation (Fig 7b). In mice, missense mutations in exon 13 of *Pde6b* are reported to cause RP [4].

To confirm whether these *rd10* mutations were also present in other individuals with retinal detachment, a total of eighteen mice were genotyped and phenotyped. Of these, eight had retinal detachment and carried the *Pde6* ^rd10/rd10 mutation. The other ten individuals that did not develop retinal detachment had heterozygous mutations (S3 Table).

## Discussion

This study identified a subset of rd6 mice that developed non-rhegmatogenous retinal detachment and severe outer retinal degeneration. Retinal detachment was observed by 6 weeks of age, with significant fundoscopic abnormalities, particularly pale vessels, distinguishing rd6-RD from standard rd6 mice.

Previous studies have linked *Mfrp* mutations to extreme hyperopia in both humans and mice [8]. In humans, MFRP deficiency is associated with shortened axial length (15.4–16.3 mm, compared to a population average of 23.5 mm), consistent with the role of the RPE in the regulation of ocular growth [14]. In normal mice, the eyeballs exhibit rapid growth involving elongation of the axis length between 3 and 6 weeks of age [15]. However, it is possible that the eyes of rd6-RD mice fail to undergo this development, resulting in localized slippage which may harm the vulnerable outer layer and promote retinal detachment. Although axial length was not measured in the present study, future investigations should specifically address this factor.

Previous studies have demonstrated that *Mfrp* deficiency impairs the development and maintenance of photoreceptor outer segments via RPE dysfunction [16] and causes RPE atrophy and consequent photoreceptor degeneration [17]. Moreover, recent biochemical analysis revealed that *Mfrp* functions as a molecular hub in the apical membrane of RPE cells, coordinating lipid homeostasis and protein trafficking [18].

Genetic analysis of rd10 mice has shown that a missense mutation in exon 13 of *Pde6b* disrupts rod photoreceptor function, leading to retinal degeneration [19]. Histologically, rd10 mice exhibit photoreceptor degeneration by 3 weeks and near-complete outer retinal loss by 7 weeks. Previous reviews have detailed the mechanisms underlying phototransduction failure and photoreceptor cell death in rd10 mice [20] and recent comparative studies have demonstrated that the *rd10* mutation induces protein mislocalization and instability, leading to cGMP-mediated cell death [21]. In this study, rd6-RD mice were found to harbor both the *Mfrp* mutation, which is associated with impaired RPE function and defective photoreceptor outer segment phagocytosis, and the *Pde6b* mutation, which causes rapid photoreceptor degeneration due to impaired phototransduction. The combination of these genetic defects may lead to marked structural instability of the outer retina, disruption of the outer limiting membrane, and weakening of adhesion between the neural retina and the RPE. Moreover, accelerated photoreceptor cell death in rd6-RD mice could promote extracellular matrix remodeling and microglial activation, further compromising retinal integrity. These molecular and cellular alterations may act synergistically to facilitate the development of non-rhegmatogenous retinal detachment in this model.

According to previous transcriptome analyses and public expression databases [22], both *Mfrp* and *Pde6b* exhibit retina-enriched expression, although they are predominantly localized to different retinal cell types. MFRP is expressed mainly in the RPE and ciliary body [16], whereas PDE6B is expressed specifically in rod photoreceptors, where it plays an essential role in phototransduction [20]. Despite these differences in cell-type localization, transcriptome studies of retinal degenerative models have reported that loss-of-function mutations in either gene can lead to downstream dysregulation of common pathways, including photoreceptor survival, oxidative stress responses, and inflammation-related signaling [23,24]. Therefore, while the upstream expression sites differ, there is overlap in the secondary gene expression changes observed in degenerating retina due to MFRP or PDE6B dysfunction.

Fundoscopic imaging of rd10 mice has also demonstrated retinal vascular sclerosis at 4 weeks, a feature also observed in rd6-RD mice. ERG responses in rd10 mice consistently decline sharply by 3 weeks and disappear by 2 months, paralleling the rapid loss of ERG signals in rd6-RD by 5 weeks.

Finally, genetic screening revealed that rd6-RD mice harbored the *Pde6b^rd10^* mutation, which has not previously been reported in standard rd6 strains. Notably, commercial suppliers such as Jackson Laboratory frequently distribute mouse strains carrying spontaneous mutations (e.g., *Pde6b^rd1^*, *Crb1^rd8^*, *Gnat2^cpfl3^*) [25], highlighting the importance of genetic verification when utilizing these models.

## Supporting information

**S1 Fig. Mean coverage by whole-genome sequencing of eight mice.** Genomic fractions analyzed by the indicated coverage are shown by colors.
(PDF)

**S2 Fig. Raw images of OCT.**
(PDF)

**S1 Table. List of primers used for PCR amplification prior to deep sequencing.**
(PDF)

**S2 Table. List of values used to build graphs.**
(XLSX)

**S3 Table. Confirming phenotype-genotype correlation in another cohort of rd6 mice used in this study.**
(XLSX)

**S1 File. Supplementary methods.**
(DOCX)

## Acknowledgments

We are deeply grateful to Dr. Shuhei Kameya for his generous support, expert advice, and extensive knowledge regarding the rd6 mouse, which greatly contributed to this study.

## Author contributions

**Conceptualization:** Asaka Lee Shiozawa, Tsutomu Igarashi.

**Data curation:** Asaka Lee Shiozawa, Maika Hosoi Kobayashi, Yusuke Shiozawa, Yoshitaka Miyagawa, Tsutomu Igarashi.

**Formal analysis:** Yusuke Shiozawa, Yasuhiro Ikeda, Fumiki Okamoto, Mashito Sakai, Takashi Okada.

**Funding acquisition:** Takashi Okada, Tsutomu Igarashi.

**Investigation:** Asaka Lee Shiozawa.

**Project administration:** Maika Hosoi Kobayashi, Yoshitaka Miyagawa.

**Supervision:** Mashito Sakai.

**Writing – original draft:** Asaka Lee Shiozawa, Tsutomu Igarashi.

**Writing – review & editing:** Yasuhiro Ikeda, Yoshitaka Miyagawa, Fumiki Okamoto, Mashito Sakai, Takashi Okada, Tsutomu Igarashi.

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
