## [Decision Letter · Decision Letter 0]

1 Jul 2025

PONE-D-25-22608Identification of coexisting Mfrprd6 and Pde6brd10 mutations causing spontaneous retinal detachment in commercially available rd6 micePLOS ONE

Dear Dr. Igarashi,

Thank you for submitting your manuscript to PLOS ONE. After careful consideration, we feel that it has merit but does not fully meet PLOS ONE’s publication criteria as it currently stands. Therefore, we invite you to submit a revised version of the manuscript that addresses the points raised during the review process.

We look forward to receiving your revised manuscript.

Kind regards,

Tatsuya Inoue

Academic Editor

PLOS ONE

Journal Requirements:

4. We note that your Data Availability Statement is currently as follows: All relevant data are within the manuscript and in Supporting Information files.

Reviewers' comments:

Reviewer's Responses to Questions

**Comments to the Author**

1. Is the manuscript technically sound, and do the data support the conclusions?

Reviewer #1: Yes

Reviewer #2: Yes

2. Has the statistical analysis been performed appropriately and rigorously? 

Reviewer #1: Yes

Reviewer #2: Yes

3. Have the authors made all data underlying the findings in their manuscript fully available?

Reviewer #1: Yes

Reviewer #2: Yes

4. Is the manuscript presented in an intelligible fashion and written in standard English?

Reviewer #1: Yes

Reviewer #2: Yes

5. Review Comments to the Author

Reviewer #1: This study provides valuable insights, but there are concerns regarding the results.

The results section does not directly mention electroretinography (ERG) or histopathology in rd10 mice, but the discussion section does mention ERG and histopathology in rd10 mice. Based on the results of whole genome sequencing, etc., does this mean that all rd6-RD mice have the Pde6 rd10/rd10 mutation?

Reviewer #2: When working with spontaneously occurring animal models, it is essential to remain alert to the possibility of additional pathogenic mutations beyond those already reported. This manuscript underscores the previously unrecognized presence of the rd10 mutation in commercially available rd6 mice, delivering an important cautionary message to researchers who rely on this model. Both the experimental design and the overall presentation are clear and persuasive.

My primary concern is that the conclusion—that severe retinal degeneration and retinal detachment in the rd6RD line arise from a combination of rd6 and rd10 mutations—is based solely on short-read whole-genome sequencing (WGS) data from a single mouse family. Because the Pde6b (c.1678C>T, p.R560C) mutation can be readily verified by Sanger sequencing, expanding the analysis to a larger cohort of animals would greatly strengthen the evidence. In addition, since rd10 mice are commercially available, crossing them with rd6 mice lacking the Pde6b mutation would test whether the rd6RD phenotype is truly reproducible, further enhancing the robustness of the conclusions.

6. PLOS authors have the option to publish the peer review history of their article (what does this mean? ). If published, this will include your full peer review and any attached files.

**Do you want your identity to be public for this peer review?** For information about this choice, including consent withdrawal, please see our Privacy Policy .

Reviewer #1: No

Reviewer #2: No

---

## [Author Response · Author response to Decision Letter 1]

9 Jul 2025

Journal Requirements:

Response: According to manuscript body formatting guidelines, we checked and created our manuscript.

Response: According to the experiments involving animals, we added the sentence. (1) methods of sacrifice; We added “Exsanguination was performed under deep anesthesia induced by ketamine/xylazine as the method of euthanasia.” in Animal husbandry. (2) methods of anesthesia and/or analgesia; We added “Mice were treated with an intraperitoneal administration of a combination of ketamine (Daiichi Sankyo Co., Ltd., Tokyo, Japan) and xylazine (Bayer Medical, Ltd., Tokyo, Japan) at 100 and 10 mg/kg body weight, respectively.” in Animal husbandry. (3) efforts to alleviate suffering; To ensure ethical approval and adherence to guidelines, the approval number from the institutional animal care and use committee and the name of the applicable guidelines are provided in the Animal husbandry section.

Response: All funding-related information has been deleted from the manuscript text.

4. We note that your Data Availability Statement is currently as follows: All relevant data are within the manuscript and in Supporting Information files.

Response: The values used to generate the graphs in Fig. 3b, Fig. 5c, and Fig. 6c are summarized in a list (S2 Table). We have added a title to S2 Table in the Supporting Information.

Response: We have prepared a file containing the raw data used for Figure 3a (as well as Figures 1 and 2), which is provided as S2 Fig. A title for S2 Fig has been added in the Supporting Information.

Response: We have corrected the reference list and confirmed that it is complete and accurate.

Reviewer #1: This study provides valuable insights, but there are concerns regarding the results.

The results section does not directly mention electroretinography (ERG) or histopathology in rd10 mice, but the discussion section does mention ERG and histopathology in rd10 mice. Based on the results of whole genome sequencing, etc., does this mean that all rd6-RD mice have the Pde6 rd10/rd10 mutation?

Response: Thank you for your question. In this study, we did not perform analyses on rd10 mice; therefore, we do not have ERG or histopathological results for rd10 mice. In the Discussion, we referred to rd10 because we identified the rd10 mutation through whole-genome sequencing. The statement means that all rd6-RD mice have the Pde6 rd10/rd10 mutation.

Reviewer #2: When working with spontaneously occurring animal models, it is essential to remain alert to the possibility of additional pathogenic mutations beyond those already reported. This manuscript underscores the previously unrecognized presence of the rd10 mutation in commercially available rd6 mice, delivering an important cautionary message to researchers who rely on this model. Both the experimental design and the overall presentation are clear and persuasive.

My primary concern is that the conclusion—that severe retinal degeneration and retinal detachment in the rd6RD line arise from a combination of rd6 and rd10 mutations—is based solely on short-read whole-genome sequencing (WGS) data from a single mouse family. Because the Pde6b (c.1678C>T, p.R560C) mutation can be readily verified by Sanger sequencing, expanding the analysis to a larger cohort of animals would greatly strengthen the evidence. In addition, since rd10 mice are commercially available, crossing them with rd6 mice lacking the Pde6b mutation would test whether the rd6RD phenotype is truly reproducible, further enhancing the robustness of the conclusions.

Response: Thank you for your question. We have prepared S3 Table to address the comment regarding expanding the analysis to a larger cohort of animals. As Supplementary Information, we have included a table summarizing the correlation between phenotype and genotype. While the original submission included 10 animals, we have added additional data to bring the total to 18 animals. Accordingly, we revised the last four sentences of the Results section and added the title for S3 Fig. in the Supporting Information.

As you pointed out, since rd10 mice are commercially available, crossing them with rd6 mice lacking the Pde6b mutation would allow us to test whether the rd6RD phenotype is truly reproducible, thereby further strengthening the robustness of our conclusions. We would like to consider this in our future studies.

---

## [Decision Letter · Decision Letter 1]

5 Aug 2025

PONE-D-25-22608R1Identification of coexisting Mfrprd6 and Pde6brd10 mutations causing spontaneous retinal detachment in commercially available rd6 micePLOS ONE

Dear Dr. Igarashi,

Thank you for submitting your manuscript to PLOS ONE. After careful consideration, we feel that it has merit but does not fully meet PLOS ONE’s publication criteria as it currently stands. Therefore, we invite you to submit a revised version of the manuscript that addresses the points raised during the review process.

We look forward to receiving your revised manuscript.

Kind regards,

Tatsuya Inoue

Academic Editor

PLOS ONE

Journal Requirements:

Additional Editor Comments (if provided):

In the present study, the authors demonstrated the phenotype-genotype correlation in the rd6-RD mice. The manuscript is well-written, however the authors should address my concerns before publication.

It seems that the authors did not mention why the retinal detachment occurred in rd6-RD mice. It would be better to add the molecular biological consideration in the Discussion section.

Is there any similarity in gene expression between Mfrp and Pde6b?

Reviewers' comments:

Reviewer's Responses to Questions

**Comments to the Author**

1. If the authors have adequately addressed your comments raised in a previous round of review and you feel that this manuscript is now acceptable for publication, you may indicate that here to bypass the “Comments to the Author” section, enter your conflict of interest statement in the “Confidential to Editor” section, and submit your "Accept" recommendation.

Reviewer #2: All comments have been addressed

2. Is the manuscript technically sound, and do the data support the conclusions?

Reviewer #2: Yes

3. Has the statistical analysis been performed appropriately and rigorously? 

Reviewer #2: Yes

4. Have the authors made all data underlying the findings in their manuscript fully available?

Reviewer #2: Yes

5. Is the manuscript presented in an intelligible fashion and written in standard English?

Reviewer #2: Yes

6. Review Comments to the Author

Reviewer #2: (No Response)

7. PLOS authors have the option to publish the peer review history of their article (what does this mean? ). If published, this will include your full peer review and any attached files.

**Do you want your identity to be public for this peer review?** For information about this choice, including consent withdrawal, please see our Privacy Policy .

Reviewer #2: No

---

## [Author Response · Author response to Decision Letter 2]

15 Aug 2025

Journal Requirements:

In the present study, the authors demonstrated the phenotype-genotype correlation in the rd6-RD mice. The manuscript is well-written, however the authors should address my concerns before publication.

1. It seems that the authors did not mention why the retinal detachment occurred in rd6-RD mice. It would be better to add the molecular biological consideration in the Discussion section.

Response: We thank the reviewer for this valuable comment. We agree that a molecular biological consideration regarding the occurrence of retinal detachment in rd6-RD mice should be included. We have now expanded the Discussion section to address this point on Page 16 and 17.

Previous studies have demonstrated that Mfrp deficiency impairs the development and maintenance of photoreceptor outer segments via RPE dysfunction [16] and causes RPE atrophy and consequent photoreceptor degeneration [17]. Moreover, recent biochemical analysis revealed that Mfrp functions as a molecular hub in the apical membrane of RPE cells, coordinating lipid homeostasis and protein trafficking [18].

Previous reviews have detailed the mechanisms underlying phototransduction failure and photoreceptor cell death in rd10 mice [20] and recent comparative studies have demonstrated that the rd10 mutation induces protein mislocalization and instability, leading to cGMP-mediated cell death [21]. In this study, rd6-RD mice were found to harbor both the Mfrp mutation, which is associated with impaired RPE function and defective photoreceptor outer segment phagocytosis, and the Pde6b mutation, which causes rapid photoreceptor degeneration due to impaired phototransduction. The combination of these genetic defects may lead to marked structural instability of the outer retina, disruption of the outer limiting membrane, and weakening of adhesion between the neural retina and the RPE. Moreover, accelerated photoreceptor cell death in rd6-RD mice could promote extracellular matrix remodeling and microglial activation, further compromising retinal integrity. These molecular and cellular alterations may act synergistically to facilitate the development of non-rhegmatogenous retinal detachment in this model.

2. Is there any similarity in gene expression between Mfrp and Pde6b?

Response: We appreciate the reviewer’s question regarding potential similarities in the expression profiles of Mfrp and Pde6b. We added the Discussion section to address this point on Page 17.

According to previous transcriptome analyses and public expression databases [22], both Mfrp and Pde6b exhibit retina-enriched expression, although they are predominantly localized to different retinal cell types. Mfrp is expressed mainly in the RPE and ciliary body [16], whereas Pde6b is expressed specifically in rod photoreceptors, where it plays an essential role in phototransduction [20]. Despite these differences in cell-type localization, transcriptome studies of retinal degenerative models have reported that loss-of-function mutations in either gene can lead to downstream dysregulation of common pathways, including photoreceptor survival, oxidative stress responses, and inflammation-related signaling [23, 24]. Therefore, while the upstream expression sites differ, there is overlap in the secondary gene expression changes observed in degenerating retina due to Mfrp or Pde6b dysfunction.

---

## [Editor Report · Decision Letter 2]

31 Aug 2025

Identification of coexisting Mfrprd6 and Pde6brd10 mutations causing spontaneous retinal detachment in commercially available rd6 mice

PONE-D-25-22608R2

Dear Dr. Igarashi,

We’re pleased to inform you that your manuscript has been judged scientifically suitable for publication and will be formally accepted for publication once it meets all outstanding technical requirements.

Kind regards,

Tatsuya Inoue

Academic Editor

PLOS ONE
---

## [Editor Report · Acceptance letter]

PONE-D-25-22608R2

PLOS ONE

Dear Dr. Igarashi,

I'm pleased to inform you that your manuscript has been deemed suitable for publication in PLOS ONE. Congratulations! Your manuscript is now being handed over to our production team.

Kind regards,

on behalf of

Dr. Tatsuya Inoue

Academic Editor

PLOS ONE